# Heterogenous Differences in Cellular Senescent Phenotypes in Pre-Eclampsia and IUGR following Quantitative Assessment of Multiple Biomarkers of Senescence

**DOI:** 10.3390/ijms24043101

**Published:** 2023-02-04

**Authors:** Samprikta Manna, Colm J. Mc Elwain, Gillian M. Maher, Marta Giralt Martín, Andrea Musumeci, Fergus P. McCarthy, Cathal McCarthy

**Affiliations:** 1Department of Obstetrics and Gynaecology, Cork University Maternity Hospital, University College Cork, T12 YE02 Cork, Ireland; 2INFANT Research Centre, University College Cork, T12 K8AF Cork, Ireland; 3Department of Pharmacology and Therapeutics, Western Gateway Building, University College Cork, T12 XF62 Cork, Ireland; 4School of Public Health, Western Gateway Building, University College Cork, T12 XF62 Cork, Ireland

**Keywords:** placental ageing, pre-eclampsia, intrauterine growth restriction, cellular senescence

## Abstract

Premature ageing of the placenta in pregnancy outcomes is associated with the persistent presence of oxidative stress and placental insufficiency reducing its functional capacity. In this study, we investigated cellular senescence phenotypes of pre-eclampsia and IUGR pregnancies by simultaneously measuring several biomarkers of senescence. Maternal plasma and placental samples were collected at term gestation from nulliparous women undergoing pre-labour elective caesarean section with pre-eclampsia without intrauterine growth restriction (PE; *n* = 5), pre-eclampsia associated with intrauterine growth restriction (n = 8), intrauterine growth restriction (IUGR < 10th centile; *n* = 6), and age-matched controls (*n* = 20). Placental absolute telomere length and senescence gene analysis was performed by RTqPCR. The expression of cyclin-dependent kinase inhibitors (p21 and p16) was determined by Western blot. Senescence-associated secretory phenotypes (SASPs) were evaluated in maternal plasma by multiplex ELISA assay. Placental expression of senescence-associated genes showed significant increases in CHEK1, PCNA, PTEN, CDKN2A, and CCNB-1 (*p* < 0.05) in pre-eclampsia, while TBX-2, PCNA, ATM, and CCNB-1 expression were evident (*p* < 0.05) and were significantly decreased in IUGR compared with controls. Placental p16 protein expression was significantly decreased in pre-eclampsia only compared with controls (*p* = 0.028). IL-6 was significantly increased in pre-eclampsia (0.54 pg/mL ± 0.271 vs. 0.3 pg/mL ± 0.102; *p* = 0.017) while IFN-γ was significantly increased in IUGR (4.6 pg/mL ± 2.2 vs. 2.17 pg/mL ± 0.8; *p* = 0.002) compared with controls. These results provide evidence of premature senescence in IUGR pregnancies, and while cell cycle checkpoint regulators are activated in pre-eclampsia, the cellular phenotype is one of cell repair and subsequent proliferation rather than progression to senescence. The heterogeneity of these cellular phenotypes highlights the complexity of characterising cellular senescence and may equally be indicative of the differing pathophysiological insults unique to each obstetric complication.

## 1. Introduction

Pre-eclampsia (PE) is a gestational hypertensive disorder which affects 3–5% of pregnancies worldwide and clinically manifests as maternal de-novo hypertension and associated maternal organ dysfunction including uteroplacental dysfunction [1]. Intrauterine growth restriction (IUGR) can be defined as a rate of foetal growth which is below the normal growth potential, i.e., between the 10th and 90th percentile as per the gestational age [2,3]. IUGR affects 13.7 million babies born yearly with low birth weight [2]. Pre-eclampsia is associated with impaired spiral artery modelling and poor placental perfusion resulting in aberrant redox signalling and an induction of oxidative stress [4]. Additionally, systemic inflammation and endothelial dysfunction are pathogenic mediators of pre-eclampsia which account for a myriad of maternal multi-organ damage and poor long-term maternal cardiovascular outcomes [4].

Senescence can be defined as a cellular defensive mechanism triggered in response to multiple external or internal stressors resulting in irreversible cell cycle arrest [5]. In conditions of exaggerated oxidative stress, persistent impairment of redox balance and subsequent DNA damage result in the initiation of a premature senescence phenotype [6]. Additionally, telomere shortening can trigger cellular senescence, but senescence is equally evident independent of telomere shortening [7,8]. Premature cellular senescence can be classified by activation of a DNA damage response pathway and its associated signaling kinases which culminate in the stimulation of cell cycle inhibitors, such as p16INK4a, p21/p53, and p27, reduction in cellular proliferation, and loss of Lamin-B activity, which is a marker of the altered enlarged and flattened morphology of senescent cells [9]. Moreover, the accumulation of mitochondrial DNA (mtDNA), as a result of ROS inducing damage and strand breaks, has been associated with premature onset of ageing in both in vivo and in vitro models, with higher susceptibility to mtDNA mutagenesis [10,11]. Activation of a senescence-associated secretory phenotype (SASP) incorporating a core set of cytokines, such as IL-6, IL-8, IL-13, IGFBPs, and MMPs, results in accumulated senile cells creating a pro-inflammatory microenvironment that indicates establishment of senescence [12]. The phenotypes of premature cellular senescence are highly heterogeneous and often differ depending on the initial stressor [13]. Senescence-specific gene expression has been extensively studied. Genes such as PTEN and CDKN2A regulate cellular senescence by acting through the p16/p19*^Arf^* pathway, whereas CHEK-1, PCNA, and CCNB-1 are modulated by DNA damage response and halt in cell cycle progression [14,15,16]. Additionally, there is no single universal biomarker that specifically identifies senescence, hence a combination of multiple markers when simultaneously present are used to confirm senescence [17].

The placenta along with maternal decidua normally exhibits progressive senescence closer to term parturition [18]. A recent review by Cox and Redman, which explored evidence of placental ageing through cellular senescence accompanied by morphological changes including modulation of pro-inflammatory phenotypes, highlighted excessive oxidative stress as a driver of premature placental ageing in adverse pregnancy conditions including pre-eclampsia [19]. This stress-induced senescence is proposed to result from poor placental perfusion, resulting in increased ROS generation that subsequently engages biochemical pathways which disrupt cell cycle progression in addition to accelerating the activation of pro-inflammatory cytokines and chemokines, which form part of the senescence-associated secretory phenotype (SASP) [13,19].

In adverse pregnancy outcomes, such as pre-eclampsia and IUGR, impaired placentation and the subsequent burden of placental oxidative stress may trigger a premature senescence phenotype leading to early parturition and pre-term delivery of the foetus [13]. Syncytiotrophoblasts show features characteristic of senescent cells, including the biomarker senescence-associated β-galactosidase (SA-β-gal), together with increased expression of the cyclin kinases inhibitors p16 and p21, and p53 [20]. To date, the association between altered telomere length and placental pathogenesis of both pre-eclampsia and IUGR has been somewhat contradictory, with some publications reporting no significant difference in placental telomere length between pre-eclampsia and healthy uncomplicated pregnancies [21,22]. Contrarily, IUGR has consistently been reported to have lower placental telomerase activity and shorter telomere length when compared with healthy uncomplicated pregnancies [23,24,25,26]. In this study, we examined multiple biomarkers of senescence in pre-eclampsia and IUGR and investigated if premature senescence phenotypes were congruent across both placental pathologies.

## 2. Results

### 2.1. Study Demographics

Thirty-nine participants all delivered by pre-labour caesarean section were recruited in Cork University Maternity Hospital between 2019 and 2021. These were grouped as: pre-eclampsia (n = 5), pre-eclampsia associated with intrauterine growth restriction (n = 8), IUGR (<10th centile) (n = 6), and maternal age-matched controls (n = 20) as shown in Table 1. There were no significant differences in maternal age, maternal BMI, or gestational age for all groups when compared with controls. While there was no significant difference in maternal mean arterial blood pressure (MAP) during the first trimester (~11 weeks) across study groups, MAP at 20–24 weeks was significantly elevated in both pre-eclampsia (96.4 ± 6.8 vs. 83.2 ± 8.4; *p* = 0.04) and pre-eclampsia associated with IUGR (97.2 ± 12.7 vs. 83.2 ± 8.4; *p* = 0.027) compared with controls. Foetal birthweight in pre-eclampsia associated with IUGR (1926 g ± 527.7 g vs. 3218 g ± 359.6 g, *p* < 0.0001) and IUGR (2585 g ± 401.8 g vs. 3218 g ± 359.6 g, *p* = 0.035) were significantly reduced when compared with controls. Similarly, foetal individualised customised centile (ICC) was significantly reduced in pre-eclampsia associated with IUGR (2.1 ± 2.5, *p* = 0.0001) and IUGR (8.11 ± 1.9, *p* = 0.0034) pregnancies compared with controls.

### 2.2. Senescence Associated Gene Expression Showed Substantial Differences in the Placental Tissue between Pre-Eclampsia and IUGR

Initially, using a Human Cellular Senescence Profiler Array with subsequent real-time PCR validation, multiple senescence-associated genes including cell cycle regulatory and DNA damage response (DDR) genes CHEK-1, CCNB-1, CDKN2A, ATM, GLB-1, ID-1, IGFBP-5, PCNA, PTEN, SOD-2, and TBX2 were examined in placental tissue samples. Of the panel of senescent-associated genes examined, there was a significant increase in placental expression of checkpoint kinase 1 (CHEK-1) (4.9-fold ± 0.94, *p* = 0.008; Figure 1A), Cyclin B (CCNB-1) (3-fold ± 1.4, *p* = 0.036; Figure 1B), proliferating cell nuclear antigen (PCNA) (6.9-fold ± 1.3, *p* = 0.013; Figure 1C), cyclin kinase CDKN2A (3.5-fold ± 0.9, *p* = 0.007; Figure 1D), and phosphatase-tensin homolog PTEN (12.8-fold ± 1, *p* = 0,003; Figure 1E) levels in pre-eclampsia. The expression of transcription factor TBX2 (0.3-fold ± 0.9, *p* = 0.016; Figure 1F), ataxia kinase ATM (0.7 ± 1.26, *p* = 0.02; Figure 1G), cyclin B CCNB-1 (0.3-fold ± 1.7, *p* = 0.028; Figure 1B), and DNA repair gene PCNA (0.2-fold ± 0.9, *p* = 0.046; Figure 1C) were significantly decreased in this IUGR, when compared with controls. There was no significant difference amongst the other senescence-associated genes between study groups (Appendix A).

### 2.3. Absolute Telomere Length in Placental Tissue Revealed No Significant Changes across All Subgroups

While placental absolute telomere length (aTL) was longer in pre-eclampsia (9.05 kbp ± 5.7 vs. 5.9 kbp ± 4.5) and pre-eclampsia associated with IUGR (6.84 kbp ± 3.2 vs. 5.9 kbp ± 4.5) compared with controls, this increase was not statistically significant (Figure 2). Adjusting for the variables maternal age, maternal BMI, gestational age, foetal individualised customised centile (ICC), and foetal sex, pre-eclampsia had a 4.88 kbp (95% CI: −0.391, 10.15) longer telomere length; additionally, pre-eclampsia associated with IUGR had a 1.65 kbp (95% CI: −3.53, 6.88) increase in telomere length compared with controls (Table 2).

When absolute telomere length was examined in IUGR (4.96 kbp ± 3.5 vs. 5.9 kbp ± 4.5), there was no significant change compared with controls (Figure 3). Adjusting for the variables maternal age, maternal BMI, gestational age, foetal individualised customised centile (ICC), and foetal sex, placental aTL was shorter in IUGR compared with controls but this reduction in aTL was not statistically significant (−1.5 kbp 95% CI: −6.852, 3.807) (Table 3).

### 2.4. Placental Expression of Cyclin-Dependent Kinase Inhibitor p16 Is Decreased in Pre-Eclampsia Only

Cell cycle arrest resulting from upregulation of cyclin-dependent kinase inhibitors, p21 and p16, is a classical feature of senescence. p16 protein expression was significantly decreased in pre-eclampsia placentas when compared with controls (*p* = 0.031) (Figure 4A), while this reduction in p16 expression was not evident in placental tissue from pre-eclampsia associated with IUGR compared with controls (Figure 4B). Additionally, there was no significant difference in p16 protein expression in IUGR (Figure 4C) when compared with controls.

There was no significant difference in p21 protein expression in pre-eclampsia (Figure 5A) and pre-eclampsia associated with IUGR (Figure 5B) when compared with controls. Similarly, there was no significant difference in p21 protein expression in IUGR (Figure 5C) when compared with controls.

A reduction in Lamin B1 protein expression has been detected in senescent cells and is induced by various triggers including cell replication and oncogene activation. Lamin B1 has roles in DNA replication and cell cycle progression in addition to maintaining nuclear envelope integrity. Expression of Lamin B1 was not significantly different in pre-eclampsia groups (Appendix A) or IUGR (Appendix A), respectively, when compared with controls. We compared absolute mtDNA copy numbers in placental tissue for pre-eclampsia subgroups and IUGR, comparing them to controls (Appendix A). Even though all groups showed increases in placental mtDNA copy numbers, both pre-eclampsia subgroups and IUGR failed to reach statistical significance compared with controls.

### 2.5. Senescence Associated Secretory Phenotypes in Maternal Plasma Reveal Different Cytokine Profiles in Pre-Eclampsia and IUGR

Senescence-associated secretory phenotypes (SASP) consist of a core panel of inflammatory cytokines and mediators including, interferon-γ (INF-γ), interleukins (IL-13, IL-6, IL-8), monocyte chemoattractant protein-1 (MCP-1) and macrophage inflammatory protein 1 alpha (MIP-1α), and matrix metalloproteinase-3 (MMP-3) [27,28,29]. Of the original panel of seven SASP markers, IL-6 was significantly increased in pre-eclampsia only when compared with controls (0.59 pg/mL ± 0.2 vs. 0.29 pg/mL ± 0.1; *p* = 0.017) (Figure 6A). Expression of the remaining six SASP markers did not differ significantly between pre-eclampsia and controls (Figure 6B–G). There was no significant difference in the expression of any SASP marker between PE/IUGR when compared with pre-eclampsia or when compared with controls. Interferon-γ expression was significantly increased (4.6 pg/mL ± 2.2 vs. 2.17 pg/mL ± 0.8; *p* = 0.002) (Figure 7E) in IUGR when compared with controls. There were no other significant changes observed in SASP markers compared with controls (Figure 7B–G).

## 3. Discussion

Premature placental ageing has been associated with adverse pregnancy outcomes such as pre-eclampsia, intrauterine growth restriction, and stillbirth. High oxidative stress and an inadequate antioxidant defense lends to impaired redox signalling and oxidative damage [30]. While the physiological triggers of senescence in vivo are not fully understood, evidence of hypoperfusion coupled with high oxidative stress and placental insufficiency may orchestrate a multistep cellular senescence response. This study sought to assess the evidence of placental ageing resulting from cellular senescence in both pre-eclampsia and IUGR by evaluating multiple biomarkers of senescence. Cellular senescence is defined as a state of terminal proliferation, accompanied by characteristic metabolic and pro-inflammatory changes. Our results showed a significant decrease in senescence-associated genes ATM, CCNB-1, PCNA, and TBX2 in IUGR placentas, and a significant increase in maternal circulating interferon-γ levels when compared with controls. On the other hand, in pre-eclampsia, we showed a significant increase in placental expression of CHEK-1, CCNB-1, PCNA, CDKN2A, and PTEN accompanied with a significant reduction in p16 placental protein levels, and a significant increase in maternal circulating IL-6 levels when compared with controls. There was no significant difference in absolute placental telomere length between any study groups, when compared with controls. The graphical abstract portrays the molecular individualities observed in PE and IUGR cohorts.

Initially, the expression of senescent-associated genes was characterised in placental tissue in pre-eclampsia, pre-eclampsia associated with IUGR, and IUGR. There was a significant increase in key regulators of DNA repair and checkpoint surveillance, namely CHEK-1 and PTEN in pre-eclampsia only, with an additional increase in expression of markers promoting proliferation (PCNA and CCNB1). Proliferating cell nuclear antigen (PCNA) is a proliferation marker, which regulates the replication complex formation and has a role in DNA metabolism, affecting DNA damage repair (DDR) pathways [31]. CHEK-1 is a DDR signalling kinase which arrests cell cycle to allow DNA repair, while the phosphatase and tensin homolog (PTEN), which had the largest increase in expression, equally controls DNA repair and regulates G1/S and G2/M checkpoints in the cell cycle to ensure appropriate cell survival. Cells experiencing DNA damage due to external or internal stressors act through the ATM/ATR pathway; CCNB-1 is downregulated leading to DNA damage-induced cellular senescence via G2 cell cycle arrest [32]. Despite the induction of the DDR, potentially facilitating DNA repair due to exaggerated oxidative stress in pre-eclampsia, the significant increase in CCNB1, a marker of G2 phase in the cell cycle, and PCNA suggest that placental cells respond to oxidative damage by promoting necessary DNA repair and re-enter the cell cycle rather than inducing senescence.

Conversely, in IUGR, there was a significant reduction in ataxia telangiectasia mutated (ATM), cell proliferation markers CCNB1, and PCNA gene expression in placental tissue compared with controls. ATM is activated by double strand break (DSB) in an attempt to repair the damage incurred [33]. TBX-2, a transcription repressor of senescence which plays an important role in the regulation of cellular fate, maintaining cellular proliferation [34,35,36,37], was also significantly reduced in IUGR only. The deficiency in the capacity to mount an appropriate DNA damage response, aligned with a significant reduction in cell proliferation, may be indicative of premature terminal differentiation of trophoblasts at a mean GA of <33 weeks, signifying a premature cellular senescence phenotype in this subgroup. While there was no significant difference in placental expression of senescence-associated genes in the pre-eclampsia associated with IUGR group, there was an obvious heterogeneity in expression of all genes, which was reflective of the individual signature profiles of pre-eclampsia and IUGR.

Lysosomal-β-galactosidase (GLB-1) has been associated as a source of senescence-associated β galactosidase (SA- β-gal) activity, a marker of cellular senescence [38]. ID-1 has been shown to act through the p16ink4a/retinoblastoma (pRb) tumour-suppressor pathway, affecting telomerase activity [39]. Insulin-like growth factor binding protein (IGBP-5) has been shown to regulate apoptosis through DDR and upregulation of the p53 pathway [40]. Superoxide dismutase 2 (SOD-2) is an antioxidant enzyme providing defense against oxidative stress [41]. There were no significant differences seen for genes GLB-1, ID-1, IGFBP-5, and SOD-2 (Appendix A).

Increased oxidative stress and accumulation of double-stranded DNA damage foci located at telomeres, irrespective of telomerase activity, may simulate premature senescence [42]. In this study, we evaluated absolute telomere length (aTL) in our study groups controlling for maternal age, BMI, gestational age, foetal sex, and foetal ICC. There was no significant difference in absolute telomere length across the three groups compared with controls. However, we showed notable variances in telomere length: pre-eclampsia with/without IUGR had longer placental aTL, whereas IUGR had a much shorter aTL. Multiple studies have failed to show any association between pre-eclampsia and placental telomere length [19,20,37]; however, IUGR has been associated with decreased placental telomerase activity and shorter telomere length when compared with healthy uncomplicated pregnancies [23,24,25,26]. Biron-Shental et al. reported shorter telomere length for pre-eclampsia and IUGR placentas compared with controls [23]. Despite the lack of evidence of significant telomere shortening in this study, it is worth noting that premature senescence can be induced independently of telomere length.

Activation of cyclin-dependent kinase (CDK) inhibitors including p16 and p21 are critical mediators of cell cycle progression and hence have important regulatory roles in promoting premature senescence. Both p21 and p16 are negative modulators of cell proliferation in part due to their control of cyclin-CDK activity in response to extracellular and intracellular signals. In this study, placental expression of p21 was not significantly different in any of the study groups compared with controls. However, placental expression of p16 was significantly reduced in pre-eclampsia only, with no change evident in pre-eclampsia associated with IUGR or IUGR alone. Conversely, placental CDKN2A gene expression was significantly increased in pre-eclampsia in our study. The CDKN2A gene locus is responsible for production of p16INK4a and p19Arf, which act as cell cycle inhibitors and are expressed at higher levels in senescent cells [15,43]; however, despite the increase in gene expression, a subsequent increase in p16 activity was not evident in pre-eclampsia. Previous studies have reported higher p16 expression in term placental syncytiotrophoblasts, equally evident in pre-eclampsia and IUGR [20,44,45,46]. The reduction of p16 expression in pre-eclampsia in this study suggests that although the evident DDR response mediated via upregulation of CHEK-1 may trigger activation of G2 checkpoint with resultant arrest of cell cycle, entry towards a senescent phenotype is not fully stabilised, which may allow cells to re-enter the cell cycle and maintain cellular mitotic property, which is supported by increase in PCNA and CCNB1 expression in pre-eclampsia [47]. As there was no change in either p21 or p16 expression in pre-eclampsia associated with IUGR or IUGR, it would suggest that growth restriction may induce the alternative senescent phenotype.

Nuclear lamins are structural proteins lining the lamina or inner nuclear envelope surface [48], which provide structural integrity. Senescent cells display enlarged flattened profiles with a distorted nuclear envelope resulting from a reduction in Lamin B1 expression. Loss of Lamin B1 has been reported as a robust marker of cellular senescence but the exact pathway initiating Lamin B1 reduction remains unclear; furthermore, loss of Lamin B1 precedes activation of the SA-β-Gal pathway and the accumulation of SASPs [49]. In this study, there was no significant difference in the placental expression of Lamin B1 in any of the study groups, which suggests that there is no overt disruption to the nuclear structure in response to DDR. Exaggerated oxidative stress in senescent cells has been associated with the accumulation of dysfunctional mitochondria. We subsequently looked for evidence of mitochondrial dysfunction by measuring mtDNA in placental tissue and reported no significant changes across the three study groups. Mitochondrial dysfunction has been associated with disease severity and complexity. Mitochondrial DNA (mtDNA) copy numbers is a measure of mtDNA levels per cell and is associated with mitochondrial enzyme activity and adenosine triphosphate (ATP) production. Mitochondria, as the source of ROS production, plays a pivotal role in the pathophysiology of placental hypoperfusion diseases as well as cellular dysfunction and ageing. Multiple publications have shown increased mtDNA in maternal circulation and cord blood in pregnancies affected by PE, PE/IUGR, and IUGR [50,51]. Contrarily, the quality of mitochondrial quality and mtDNA copy number is significantly reduced in ageing cells [52,53,54]. This study did not provide evidence of increased mtDNA copy numbers in PE, PE associated with IUGR, and IUGR, and based on this surrogate marker of mitochondrial dysfunction, we propose that mitochondrial dysfunction is not a significant contributor to the heterogeneity evident in our study.

One of the hallmarks of senescence is the ability of growth-arrested cells to remain metabolically active and secrete SASPs, which affects events within the cell and causes senile modulation of the extra-cellular environment as well as neighbouring cells [55]. Our results showed significant increase in IL-6 levels in maternal plasma in women with pre-eclampsia only. Multiple reports have shown that increased IL-6 levels are associated with both severity and onset of pre-eclampsia [56,57]. Despite its well-defined role as an inflammatory mediator, IL-6 also has a major role in cell cycle regulation through activation of the janus kinase-signal transducer and activator of transcription (JAK/STAT) and mitogen-activated protein kinase (MAPK) signal transduction pathways [34,58]. The classical MAPK/ERK family is an intercellular checkpoint for cellular mitogenesis, controlling cell cycle progression (34). In this study, the increase in IL-6 may be more indicative of systemic maternal inflammation in pre-eclampsia rather than a consequence of a senescent phenotype in pre-eclampsia. Interferon gamma (IFN-γ), a pro-inflammatory cytokine, has been reported to induce cellular senescence through the p53-dependent DNA damage signaling pathway [48,59,60]. Our results show significant elevation of circulating IFN-γ in IUGR pregnancies, which may be indicative of premature cellular senescence. Similar to all other senescent biomarkers, there was no significant difference in circulating cytokines in the pre-eclampsia associated with IUGR study group.

The approach of measuring multiple senescence markers in pre-eclampsia, pre-eclampsia associated with IUGR, and IUGR (<10th centile) is a significant strength of this study as it facilitated a complete investigation of premature senescence phenotypes in these obstetric complications. To date, identifying the physiological stressors of senescence in vivo are not fully elucidated. Despite the limitation of small sample numbers in this study, it is evident that there are significant differences in cell proliferation/senescence phenotypes in pre-eclampsia and IUGR. The heterogeneity of these phenotypes in pre-eclampsia and IUGR alone are equally different in pre-eclampsia associated with IUGR, where an altered molecular profile was evident with no obvious evidence of cellular senescence or altered proliferation.

## 4. Materials and Methods

### 4.1. Study Design

Subjects were recruited as part of the COMRADES study, a non-interventional cohort study of nulliparous singleton pregnancies with the aim of characterising placental premature senescence in pre-eclampsia (PE) and intrauterine growth restriction (IUGR). Patients were recruited from Cork University Maternity Hospital, Ireland, between 2019 and 2021. The COMRADES study was conducted according to the guidelines laid down in the Declaration of Helsinki, and all the procedures were approved by the Clinical Research Ethics Committee of the Cork Teaching (ECM4 (ff) 04/12/18), and all women provided written informed consent. After receiving consent from participants, samples were collected from nulliparous, singleton pregnancies undergoing pre-labour caesarean section. Pre-eclampsia was defined according to the ISSHP 2018 guidelines: BP ≥ 140 mm Hg systolic or ≥90 mm Hg diastolic measured on at least 2 occasions at least 4 h, associated with proteinuria (>300 mg protein on 24-h collection, urine protein creatinine > 30 mg/mmol or +3 dipstick) [1]. Pre-eclampsia participants were subdivided into two groups: associated with growth restriction (n = 8) and no growth restriction (n = 5). IUGR was defined according to RCOG as estimated foetal weight less than 10th centile for gestational age [61]. IUGR-recruited pregnancies were classified as <10th centile (n = 6). Matched selected controls (n = 20) were collected from healthy pregnant women who had uncomplicated pregnancies, which were defined as pregnancies not affected by pre-eclampsia or IUGR and delivered at >37 weeks. Controls were matched with the pre-eclampsia and IUGR cases for maternal age.

### 4.2. Sample Collection

Maternal term plasma was collected in EDTA vacutainers (BD, Ireland) and centrifuged at 2400× *g* for 10 min at 4 °C. Placental tissue was sectioned within 20 min of delivery, with 5 tissue sections (2–4 cm diameter each) taken from identified regions across the placenta. All samples were snap frozen in liquid nitrogen and stored at −80 °C [62].

### 4.3. Placental DNA Extraction and Absolute Telomere Length Quantification

Genomic DNA was extracted from 10 mg of placental tissue using the DNeasy Blood and Tissue kit (Qiagen, Dublin, Ireland) in accordance with the manufacturer’s instructions. Absolute quantification of telomere was determined using the ScienCell qPCR Kit (#8918). The telomere primer set recognizes and amplifies telomere sequences, while the single copy reference (SCR) primer set recognizes and amplifies a 100 bp-long region on human chromosome 17 and serves as a reference for data normalisation. All reactions with genomic DNA sample (5 μg/μL) sample were performed in duplicate, and template controls were included in each run. Reference genomic DNA was used as an internal control. Comparative ∆∆Cq (Quantification Cycle Value) method was used with manufacturer made standard curve for quantification using Cq (Quantification Cycle) values of telomere qPCR (TEL) and single copy reference qPCR (SCR) obtained for the genomic DNA samples as per the manufacturer’s instructions.

### 4.4. Absolute Quantification of Placental mtDNA

Mitochondrial DNA (mtDNA) extraction was done following the methodology described by Williamson et al. (2018) [50]. Real-time PCR was used for mtDNA assessment through StepOne Plus Detection system (Applied Biosciences, Waltham, MA, USA) using Taqman probes for mtDNA hMitoF5, hMitoR5 [50,63]. A standard curve was used to determine the absolute quantification of mtDNA and represented as copies/mL [63,64].

### 4.5. Isolation of RNA and Real-Time PCR Analysis

Placental tissue was mechanically homogenised in Trizol Reagent (Invitrogen™ Thermo Fisher Scientific, Dublin, Ireland) using TissueLyser II (Qiagen, Manchester, UK) and RNA was isolated which was subsequently purified using the Qiagen RNeasy^®^ Plus Mini (#74134) clean-up protocol according to the manufacturer’s instructions. cDNA synthesis was performed using the Superscript IV Double-Stranded cDNA Synthesis Kit (Invitrogen™ Thermo Fisher Scientific, Ireland). Initial assessment for senescence-associated genes were performed using the RT2 Profiler PCR Array Gene Expression for senescence-associated genes in 96-well plates (Cat no. PAHS-050Z, Qiagen, Ireland). The samples from each group were pooled into and loaded on each 96-well plate. In total, there were 4 groups: PE, PE associated with IUGR, IUGR, and controls, and 4 plates were subjected to Real Time qPCR on the Applied Biosciences StepOne Plus Detection System. The qPCR Assays used in PCR Arrays enables 96 genes in a 96-well plate to be simultaneously analysed using the RT^2^ SYBR Green q PCR Array System protocol. Relative quantification was performed using data analysis web portal (GeneGlobe, Germantown, MD, USA, https://geneglobe.qiagen.com/us/ (accessed on 17 January 2023)) to calculate the fold change using 2^−ΔΔCT^ formula, where ΔCT is calculated between gene of interest (GOI) and an average of reference genes (HKG), followed by ΔΔCT calculations. Five HKG genes, namely ACTB, B2M, GAPDH, RPLP0, and HPRT1 were included in each PCR plate as internal controls. Eleven genes were selected for validation based on the significant differences between groups when compared with controls, obtained from the RT2 profiler PCR Array: CHEK1, CCNB1, PCNA, PTEN, CDKN2A, ATM, TBX2, GLB1, ID1, IGFBP5, and SOD2. Taqman probes were used for genes of interest (Applied Biosciences, Westmeath, Ireland). The Tata box protein (TBP) gene was used as an internal control with SYBR green primers (Appendix A). Standard relative PCR quantification was performed to validate the fold change using 2^−ΔΔCT^) formula. Taqman probes were used for genes of interest (Applied Biosciences, Ireland). The Tata box protein (TBP) gene was used as an internal control (Eurofins, Dublin, Ireland; Appendix A). TBP has been shown to be most stable in placental tissue, ensuring sensitivity and specificity of gene expression profile [65]. Real-time PCR was performed on the Applied Biosciences StepOne Plus Detection System. Standard relative PCR quantification was performed using the manufacturer’s instruction, and fold change was calculated based on relative change between GOI and the internal standard.

### 4.6. Protein Isolation and Western Blot

Placental tissue (50 mg) was homogenized using TissueLyser II (Qiagen, Hilden, Germany) and protein was isolated using tetraethylammonium bicarbonate buffer (Sigma-Aldrich, Co., Wicklow, Ireland) supplemented with a phosphatase inhibitor (PhosSTOP™, Roche, Dublin, Ireland) and protease inhibitor (Roche, Ireland). PE, PE associated with IUGR, and IUGR cases were age-matched with controls. We separated 35 μg of protein by SDS-PAGE on a 10% TRIS polyacrylamide-glycine gel (Bio-Rad Laboratories, Kildare, Ireland) and transferred it to a methanol-activated PVDF membrane (Amersham™ Hybond^®^ Sigma-Aldrich, Wicklow, Ireland). Membranes were blocked in 5% BSA for 1 h at room temperature and incubated overnight at 4 °C with primary antibodies; anti-p21 (1:1000 Abcam; ab109520, Cambridge, UK), anti-Lamin B1 (1:1000 Cell Signaling; E6M5T, Cork, Ireland), anti-p16 (1:500 Abcam; ab108349, Cambridge, UK), or anti-GAPDH (1:3000 Cell Signaling #2118, Brennan and company, Dublin, Ireland), and anti- β-Tubulin (1:2000 Cell Signaling, #2146, Brenan and company, Dublin, Ireland), secondary anti-rabbit IgG, HRP-linked antibody (1:2000 Cell Signaling Technology, #7074 Brenan and company, Dublin, Ireland). GAPDH and β-Tubulin were selected as internal reference control proteins for western blots because of their stability and expression within placental cells [66,67,68]. After overnight incubation membranes were washed and incubated at room temperature for 1 h with anti-HRP secondary antibody, membranes were washed in TBS-T and developed using Pierce™ ECL Western Blotting Substrate, Thermo Fischer Scientific, Biosciences, Dublin, Ireland) and LICOR Odyssey image analyzer. Densitometry was performed using LICOR Image Studio Lite Version 5.2. After the protein of interest was developed, membranes were stripped of residual primary and secondary antibodies using a Western Blot Stripping Buffer (Thermo Fischer, Dublin, Ireland) for 30 min with gentle shaking and re-probed with a suitable housekeeper protein.

### 4.7. Senescence Associated Secretory Phenotypes (SASP) Analysis

Maternal plasma samples were analysed for SASP using the Human U-Plex kit IL-6, IL-8, IL-13, IL-1α, IFN-γ, MCP-1, and MIP-1α (MesoScale Discovery, Rockville, MD, USA; K15067L) and the Human MMP3 Ultra-sensitive kit (MesoScale Discovery, Rockville, MD, USA; K15034CL). All samples were prepared using the manufacturer’s instructions and plates analysed on the Meso QuickPlex SQ 120 platform. Results were generated as calculated concentration means on the Mesoscale (MSD) Discovery Workbench 4.0 assay analysis software. The MSD analysis software determines individual cytokine concentrations from electro chemiluminescent signals via backfitting to the calibration curve.

### 4.8. Statistical Analysis

All data are expressed as mean ± SD, or fold change relative to control. Analysis was performed using GraphPad Prism 8.3. Comparisons of data between cases and controls were performed using a one-way ANOVA. The two-tailed Wilcoxon matched-pairs signed rank test was used for statistical analysis of Figure 4 and Figure 5. *p* values < 0.05 were considered as statistically significant. Linear regression analysis, comparing absolute telomere length in cases and controls, was done using STATA 14.2.

## 5. Conclusions

In the study we showed that despite activation of DNA damage response, in part as a result of oxidative damage, the cellular phenotype signified a re-entry into the cell cycle and continued cell proliferation rather than progressing towards senescence in pre-eclampsia. IUGR presented a completely different phenotype, which was representative of significantly reduced cell proliferation and premature terminal differentiation which may progress towards senescence. Despite the assessment of multiple biomarkers of senescence in this study, it highlights the complexity in distinguishing between cellular subtypes at a single cell level as well as between cell types, which may be reflective of the differing orchestrators of placental pathogenesis of these pathologies.

### Limitations

This study has a limited sample size which might influence the overall conclusions drawn in this study. The limited sample number was due to a stringent study design of recruiting nulliparous women with PE, PE associated with IUGR, and IUGR cohorts. Only term pregnancies were included in the study to overcome the effect of premature births. Women with IUGR pregnancies that were complicated by any foetal abnormalities were excluded.

## Figures and Tables

**Figure 1 ijms-24-03101-f001:**
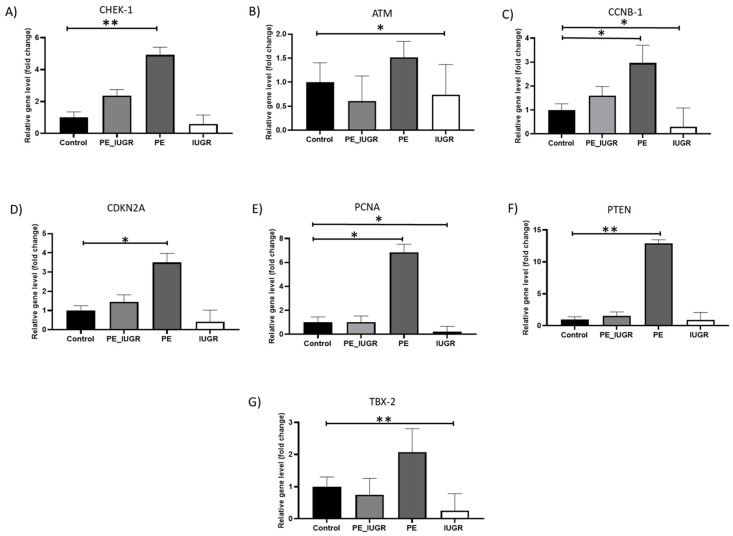
Senescence-associated gene expression in placental tissue in pre-eclampsia and IUGR. (**A**) CHEK-1 expression in pre-eclampsia only was significantly elevated (*p* = 0.006); (**B**) ATM was significantly decreased in IUGR (*p* = 0.02); (**C**) CCNB-1 was significantly elevated in pre-eclampsia only (*p* = 0.028), whereas it was significantly reduced in IUGR (*p* = 0.036); (**D**) CDKN2A was significantly elevated in pre-eclampsia only (*p* = 0.012); (**E**) PCNA was significantly elevated in pre-eclampsia only (*p* = 0.014), whereas it was significantly decreased in IUGR (*p* = 0.04); (**F**) PTEN was significantly elevated in pre-eclampsia only (*p* = 0.005); (**G**) TBX-2 was significantly reduced in IUGR only (*p* = 0.005). All measurements were relative gene expression compared with controls. Data are mean fold change compared with controls at level 1 ± SD, *p* < 0.05; *p*-value < 0.05, it is flagged with one star (*) and *p*-value < 0.01, it is flagged with two stars (**).

**Figure 2 ijms-24-03101-f002:**
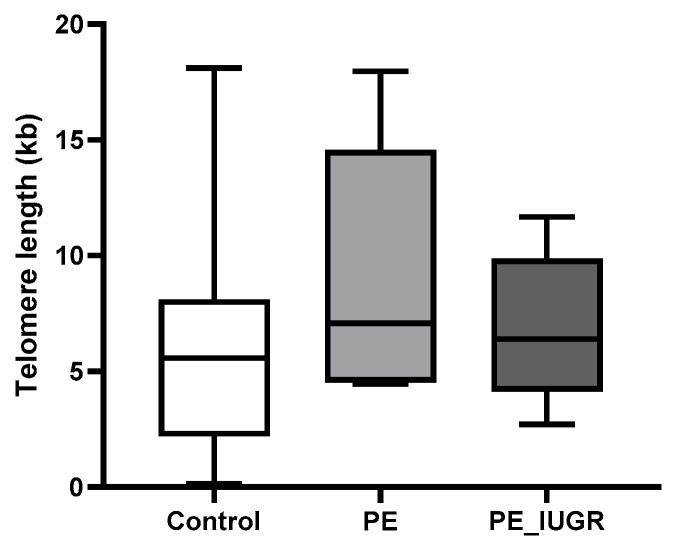
Absolute telomere length in pre-eclampsia. aTL in placental tissue was not significantly altered in pre-eclampsia or pre-eclampsia associated with IUGR compared with controls. Data are represented as mean aTL ± SD, *p* < 0.05.

**Figure 3 ijms-24-03101-f003:**
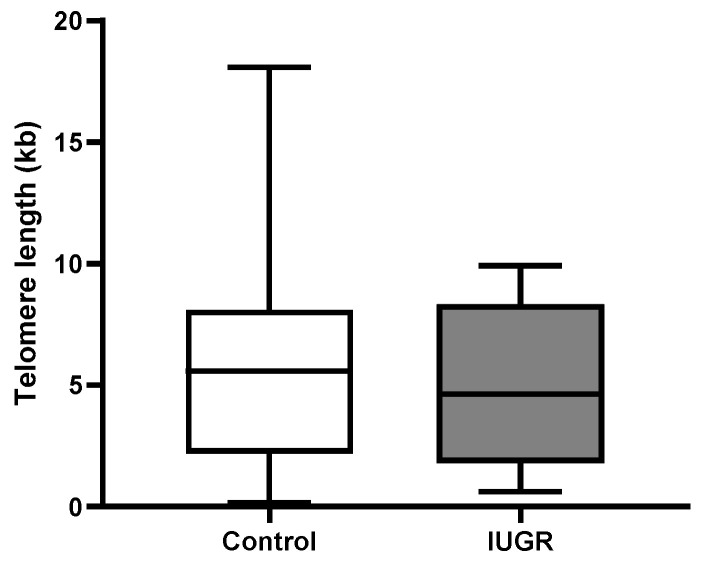
Absolute telomere length in IUGR. ATL in placental tissue showed no significant changes in IUGR compared with controls. Data are represented as mean aTL ± SD, *p* < 0.05.

**Figure 4 ijms-24-03101-f004:**
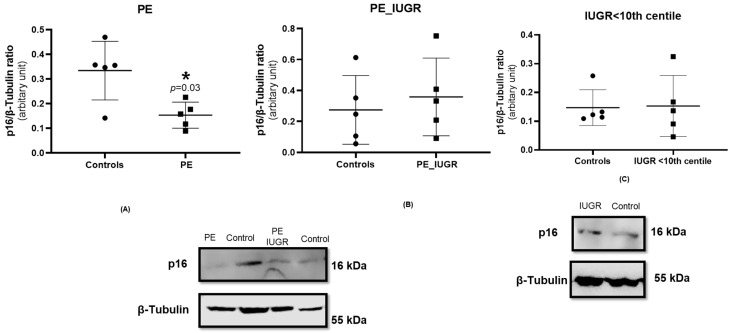
Western blot for placental p16 protein expression performed using the standard SDS-page gel method for (**A**) pre-eclampsia only, n = 5, (**B**) pre-eclampsia associated with IUGR, n = 5, and (**C**) IUGR, n = 5. Statistical analysis with Wilcoxon matched-pairs signed rank test with two-tailed test, * *p* < 0.05 vs. control placenta. *p*-value < 0.05, it is flagged with one star (*). Protein expression of p16 was normalised to the expression of the housekeeping reference protein β-tubulin.

**Figure 5 ijms-24-03101-f005:**
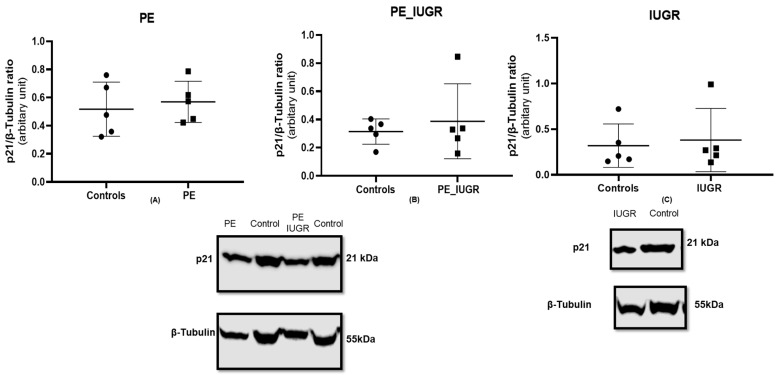
p21 expression in placental tissue using the standard SDS-page gel method; (**A**) pre-eclampsia, n = 5, (**B**) pre-eclampsia associated with IUGR, n = 5, and (**C**) IUGR >10th centile, n = 5. Statistical analysis with Wilcoxon matched-pairs signed rank test with two-tailed test. Protein expression of p21 was normalised to the expression of the housekeeping reference protein B-tubulin.

**Figure 6 ijms-24-03101-f006:**
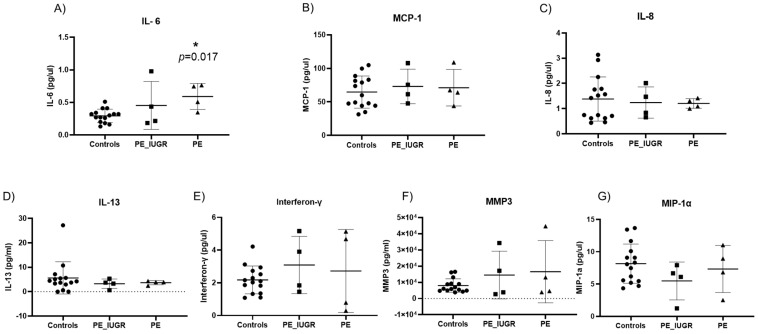
Senescence-associated secretory phenotypes (SASP) in pre-eclampsia with or without intrauterine growth restriction. (**A**) IL-6 was significantly increased in maternal plasma of pre-eclampsia only pregnancies (*p* = 0.017) when compared with controls. All other cytokines: (**B**) MCP-1, (**C**) IL-8, (**D**) IL-13, (**E**) Interferon-γ, (**F**) MMP-3, and (**G**) MIP-1α showed no significant differences between pre-eclampsia subgroups compared with controls. Data are represented as mean ± SD, *p* < 0.05; *p*-value < 0.05, it is flagged with one star (*).

**Figure 7 ijms-24-03101-f007:**
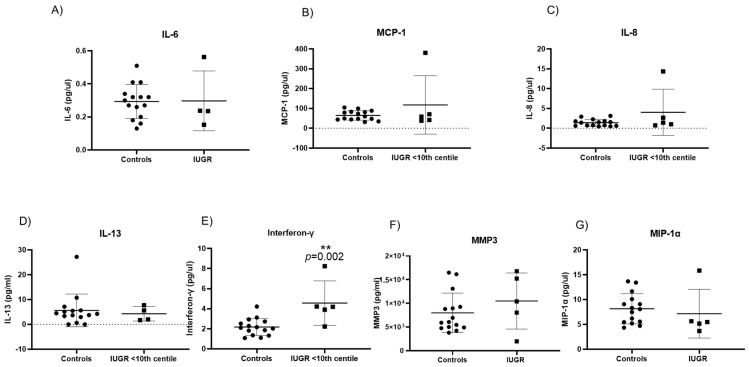
Senescence-associated secretory phenotypes (SASP) evident in intrauterine growth restriction IUGR: (**E**) Interferon-γ (*p* = 0.002) of IUGR pregnancies were significantly increased in maternal plasma when compared with controls. All other cytokines: (**A**) IL-6, (**B**) MCP-1, (**C**) IL-8, (**D**) IL-13, (**F**) MMP-3, and (**G**) MIP-1α showed no significant difference when compared with controls. Data are represented as mean ± SD, *p* < 0.05; *p* = 0.002 flagged as two stars (**).

**Table 1 ijms-24-03101-t001:** Patient characteristics in study cohort. Data are represented as mean ± SD. Mean arterial pressure has been calculated as MAP = [Systolic BP + (2 × Diastolic BP)]/3.

	Pre-Eclampsia	Pre-Eclampsia Associated with IUGR	IUGR (>10th Centile)	Term Controls
	(n = 5)	(n = 8)	(n = 6)	(n = 20)
Maternal				
Mean Age (years ± SD)	38.6 ± 5.2	33.35 ± 3.1	33.67 ± 3.1	33.7 ± 4.40
Mean BMI (kg/m^2^ ± SD)	29.9 ± 9.6	27.7 ± 3.7	25.31 ± 3.5	24.4 ± 5.4
Mean Arterial Blood Pressure (mmHg) ± SD	92 ± 10.4	77.08 ± 32.4	84.61 ± 5.2	86.3 ± 8.9
(10–12 weeks)				
Mean Arterial Blood Pressure (mmHG) ± SD	96.4 ± 6.8 *	97.2 ± 12.7 *	83.1 ± 7.2	83.2 ± 8.4
(20–24 weeks)				
Foetal				
Mean Gestational Age at delivery	36 ± 1.9	34.6 ± 2.7	32.43 ± 13.1	38 ± 0.7
(weeks ± SD)				
Mean birth weight (grams ± SD)	2878 ± 788.5	1926 ± 572.7 *	2585 ± 401.8 *	3218 ± 359.6
Foetal Sex	F = 3	F = 3	F = 4	F = 10
(M = Male; F = Female)	M = 2	M = 5	M = 2	M = 10
Foetal Individualised Customised Centile	46.8 ± 39.3	2.15 ± 2.5 *	8.11 ± 1.9 *	40.4 ± 21.9
(±SD)				

* Statistically significant compared with controls with *p* < 0.05.

**Table 2 ijms-24-03101-t002:** Crude and adjusted association between placental absolute telomere length (aTL) in kbp for pre-eclampsia and pre-eclampsia associated with IUGR, compared with controls. ^a^ Adjusted for maternal age, BMI, gestational age, ICC, and foetal sex. Abbreviations: 95% CI = 95% confidence interval, BMI = body mass index, ICC = individualised customised centile, aTL = absolute telomere length.

Exposure Groups	Crude Estimate (95% CI)	Adjusted Estimate ^a^ (95% CI)
Pre-eclampsia only (N = 5)	3.191 (−1.288, 7.671)	4.882 (−0.391, 10.155)
Pre-eclampsia with IUGR (N = 8)	0.986 (−2.761, 4.734)	1.675 (−3.527, 6.878)

**Table 3 ijms-24-03101-t003:** The Crude and adjusted association between placental absolute telomere length (aTL) in kbp for IUGR placentas compared with controls. ^a^ Adjusted for maternal age, BMI, gestational age, ICC, and foetal sex. Abbreviations: 95% CI = 95% confidence interval, BMI = body mass index, ICC = individualised customised centile, aTL = absolute telomere length.

Exposure Groups	Crude Estimate (95% CI)	Adjusted Estimate ^a^ (95% CI)
IUGR < 10th centile (N = 6)	−0.895 (−5.214, 3.423)	−1.522 (−6.852, 3.807)

## Data Availability

Anonymised data that support the findings of this study are available from the corresponding author, upon reasonable request. Anonymised data will be made available in a suitable open access repository in the future.

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
