# Peer review of "Heterogenous Differences in Cellular Senescent Phenotypes in Pre-Eclampsia and IUGR following Quantitative Assessment of Multiple Biomarkers of Senescence"

_ijms, 2023, doi:10.3390/ijms24043101_

Round 1
Reviewer 1 Report
Manna et al. present the study with the title "Heterogenous differences in cellular senescent phenotypes in Pre-eclampsia and IUGR following quantitative assessment of multiple biomarkers of senescence" trying to elucidate the role of senescence in a variety of placental disorders. Overall this is an interesting study since it supports the hypothesis of early differentiation of cells in the IUGR placenta.
While overall an interesting study the following limitations need to be addressed.
- small sample number - justification missing
- tissue heterogeneity - cell population are different in the different pathologies and those need to be addressed and discussed
- the western blots are of poor quality, overexposed and high back ground
- how did the author know that GAPDH may or may not be different between the various disorders
- PCR - analysis - presentation of PCR methods unacceptable
- which house keeping genes were used and why
- Limitations of the study, data and interpretation should be included
Author Response
While overall an interesting study the following limitations need to be addressed.
- small sample number - justification missing
We agree and have acknowledged this limitation and added justification for small sample size in the Limitations section of the manuscript, as indicated in the following paragraph:
“Limitations: This study has a limited sample size which might influence the overall conclusions drawn in this study. The limited sample number was resultant of a stringent study design of recruiting nulliparous women with PE, PE associated with IUGR and IUGR cohorts. Only term pregnancies were included in the study to overcome the effect of premature births. Women with IUGR pregnancies which were complicated by any fetal abnormalities were excluded.”
- tissue heterogeneity - cell population are different in the different pathologies and those need to be addressed and discussed
The authors have accounted for tissue heterogeneity through their dissection methodology. The placenta is by nature an organ with extreme variations between individuals. Our sectioning methodology was consistent with published guidelines (Burton, G J et al. “Optimising sample collection for placental research.” Placenta vol. 35,1 (2014): 9-22. doi:10.1016/j.placenta.2013.11.005)
The following steps were undertaken by the authors:
- All placentas were collected from nulliparous singleton pregnancies undergoing Caesarean section delivery
- Five identical sections were collected from around the placenta, avoiding cord insertion region and major blood vessels.
- All membranes, including fetal membrane were removed
- The five separate sections were combined to get a global picture of the placenta
Tissue heterogeneity amongst placental pathologies were commonly seen at a cellular level, which can be only analysed through single cell transcriptomics. The authors collected the trophoblast layer from the placenta. Moreover, all samples were matched for maternal factors such as age, BMI and gestation age between cases and controls as closely as possible.
- the western blots are of poor quality, overexposed and high back ground
The authors have revised the western blots and uploaded for review.
- how did the author know that GAPDH may or may not be different between the various disorders
The authors would like to address that GAPDH, or B-Tubulin were used as internal standards for the western blot analysis. Both GAPDH and B-tubulin have been used extensively as internal controls in placental research publications. We have added references in the main manuscript to support the use of GAPH as an internal standard. The decision to use two different housekeeping proteins was dependant on the proximity of target protein being analysed. For example, B-Tubulin has a molecular mass of 55kDa which is very close to the Lamin B1 molecular mass of 68.25 kDa, hence GAPDH (molecular mass 37kDa) was used as the internal control. The authors initially performed a Western for protein of interest and then stripped the membrane and blotted for the internal control on the same membrane. The authors have added the following sentences in the methods section of the main manuscript:
GAPDH and β-Tubulin were selected as internal reference control proteins for western blots because of their stability and expression within placental cells [58-60].
- PCR - analysis - presentation of PCR methods unacceptable
The authors have added the following paragraph detailing the PCR analysis methodology. The manuscript now reads as follows:
“Initial assessment for senescence associated genes were performed using RT2 Profiler PCR Array Gene Expression for senescence associated genes in 96-wells plate (Cat no. PAHS-050Z, Qiagen, Ireland). The samples from each group were pooled into and loaded on each 96-well plates, in total we have 4 groups; PE, PE associated with IUGR, IUGR and controls; and 4 plates were subjected to Real Time qPCR on the Applied Biosciences StepOne Plus Detection System. The qPCR Assays used in PCR Arrays enables 96 genes in a 96-well plate to be analysed simultaneously using the RT² SYBR Green q PCR Array System protocol. Relative quantification was performed using data analysis web portal (GeneGlobe, USA, https://geneglobe.qiagen.com/us/) to calculate the fold change using using 2^ (-ΔΔCT) formula, where ΔCT is calculated between gene of interest (GOI) and an average of reference genes (HKG), followed by ΔΔCT calculations. Five HKG genes namely ACTB, B2M, GAPDH, RPLP0 and HPRT1 were included in each PCR plate as internal controls.
11 genes were selected for validation based on the significant differences between groups when compared to control, obtained from the RT2 profiler PCR Array: CHEK1, CCNB1, PCNA, PTEN, CDKN2A, ATM, TBX2, GLB1, ID1, IGFBP5 and SOD2. Taqman probes were used for genes of interest (Applied Biosciences, Ireland). The Tata Box Protein (TBP) gene was used as an internal control with SYBR green primers (Supplementary table 1).
Standard relative PCR quantification was performed to validate the fold change using 2^ (-ΔΔCT) formula.
Supplementary Table 1:
TBP Forward Sequence TGTATCCACAGTGAATCTTGGTTG
TBP Reverse Sequence GGTTCGTGGCTCTCTTATCCTC
- which house keeping genes were used and why
TBP (TATA box binding protein was used as the internal control gene. The authors have added the following in methods section. The manuscript now reads as follows:
“The Tata Box Protein (TBP) gene was used as an internal control (Eurofins, Ireland; Supplementary Table 1). TBP has been shown to be most stable in placental tissue, ensuring sensitivity and specificity of gene expression profile [61].”
- Limitations of the study, data and interpretation should be included
We have further acknowledged the limitations of our research as per response number 1 and adjusted the manuscript as described.
- Data Availability: Anonymized data that support the findings of this study are available from the corresponding author, upon reasonable request. Anonymized data will be made available in a suitable open access repository in the future.”

Reviewer 2 Report
The article of McCarthy and colleagues is devoted to the interesting problem of dissecting the mechanism of occurrence of various pathological manifestations of initial placental insufficiency in preeclampsia and/or IUGR. This problem is known and interesting because in both cases, oxidative stress in the placenta is involved in the development of the pathology, and the consequences are different - either just preeclampsia, or IUGR, or PE-associated IUGR. The issue of placental aging associated with the accumulation of oxidative damage has not yet been resolved, since data on aging at the morphological, molecular and functional levels are contradictory, despite the many studies in this area performed in the last decade. The authors set the task of showing whether cell aging, accompanied by shortening of telomeres, really determines the path of pathology development to varying degrees, or is it a matter of functional limitation through cell cycle inhibition?
Despite the fact that the authors obtained unequivocal data in favor of the second hypothesis (leave a small sample out of the brackets, the authors themselves understand that it would be nice to validate the study on an expanded sample), several lines of logic are mixed in the discussion of the results, which does not draw convincing conclusions.
For example, for what purpose the Аuthors in the chapter show the results that the absolute length of telomeres does not differ from the control anywhere (let's leave aside static tricks about values adjusted to age, etc. - this is unjustified for such small samples), and in the discussion Authors unexpectedly conclude –« Our results showed a significant decrease in senescence associated genes ATM, CCNB-1, PCNA and TBX2 in IUGR placentas, accompanied with shorter absolute placental telomere length and significant increase in maternal circulating Interferon-γ levels when compared to controls; whereas in pre-eclampsia, we showed a significant increase in placental expression of CHEK-1, CCNB-1, 264 PCNA, CDKN2A and PTEN, a significant reduction in p16 placental protein levels, extended absolute placental telomere length and a significant increase in maternal circulating IL-6 levels when compared to controls.
This is all the more incomprehensible, since the absence of a difference in telomere length against the background of functional shutdown of the cell cycle in IUGR is fully consistent with the Authors' concept of different paths for the development of a pathologyу from placental insufficiency to preeclampsia or IUGR.
What is the purpose of suddenly and without regard to the paradigm of the article mentioning the same content of mtDNA in the placenta in all groups and making a conclusion about the absence of mitochondrial dysfunction? The number of mtDNA copies is in no way related to the quality of mitochondria, and it is incorrect to speak of their dysfunction (rather than quantity) based on the definition of this parameter.
Taking into account other shortcomings listed below, the article should be finalized for subsequent publication, since the new data presented in it and the differences in the mechanism of pathology formation in PE and IUGR are of great importance for appropriate further research and possible clinical recommendations, and the contribution of the Authors is undoubted.
Questions and recommendations:
Abstract and introduction
Although the Abstract lists the abbreviations of proteins, the content of which changes in the placenta and plasma during PE and IUGR in comparison with the control, nothing is deciphered in the Introduction (and even these proteins are not mentioned) - what are these proteins and why it is interesting to study these proteins, there is no logic, leading to the formulation of the problem. Again, why is mtDNA mentioned in terms of quantification and where is it supposed to see the significance of changes in the amount of mtDNA, especially as a result of damage (increase?) - in tissue, in plasma, in the intercellular space? "Moreover, the accumulation of mitochondrial DNA (mtDNA) as a result of ROS inducing damage and strand breaks..." - Probably, did the authors mean the accumulation of mtDNA MUTATIONS as a result of ROS? But further nowhere is it said about mutations and they were not defined. We also note that the vision of the Authors in the style of "...result from poor placental perfusion, resulting in ROS accumulation that subsequently..." is incorrect, since ROS cannot accumulate by themselves due to their high reactivity and short lifetime.
I would like to see a clearer logic here, for example (not necessarily exactly this way, I do not insist!) In the style - incomplete trophoblast invasion - permanent placental reperfusion - hypoxia / reoxygenation - oxidative stress - apoptosis and compensatory processes to restore placental function - such and such with PE, such and such with IUGR and the difference at the level of different regulation of tissue aging.
Results:
Just mistake I expect phosphatase
137 line “…Phosphotase-Tensin homolog PTEN ….”
Fig 4 – it is not clear meaning of blots there – Is it typical blots? Should be written then in the caption to the figure.
Are these differences everywhere, almost an order of magnitude, between the samples of the control group? And why do the values for the control group vary so much between experiments - medians around 0.35 for a), less than 3 for b) and about 0.1 for c)? The nature of the distribution is also very different. Is there an explanation why tubulin-normalized values change so much for the same group (n=5) depending on what they are paired with?Generally unconvincing.
The values of the medians for p21/tubulin in Fig. 5 for control samples are even more depressing - 0.5; 0.3; 2.5 for a), b), c). How can it be in principle for samples of the same group, if we are discussing relative values (normalized)? - from blot to blot, p21 and tubulin stain better, sometimes worse, and this leads to a change in their ratio? what then does the definition and comparison with PE and IUGR, PE+IUGR matter? Maybe there is the result of an artifact? This should be analyzed while related samples and appropriate statistical method. Wilcoxon, for example, not MW!
Line 225 – “Senescence Associated Secretory Phenotypes (SASP) consists of a core panel of inflammatory cytokines and mediators including…” – where is this panel came from? Ref?
A separate issue is the technique and quality of Western blots. First, it is not clear how two different signals from two proteins can be seen on the same blot (original images). Was the stripping technique used, was the overlay used in the program, why is this not described in the methods? And to my regret - Lamin B1 and GAPDH on the left and on the right are different membranes, not “same blot” - there the corners are cut off in different places and the markers dispersed in different ways on electrophoresis. Secondly, the methods do not specify which markers were used. Thirdly, in Supplementary Figure II, the band corresponding to this protein is quite clear, while in samples from the original images it is generally difficult to determine where the protein is localized, everything is so nonspecific (separation is generally unsuccessful). In general, after looking at the original images, it is not clear how such data on the content of these proteins were obtained, the blots are inconclusive.
Discussion:
Discussion of the results is a strong part of the article, although a description of the operation of individual proteins could be given in the introduction. Logical and persuasive. The only negative is long and poorly readable sentences (for example, from line 260 to line 267), but this is easy to fix.
The discussion from lines 320 to line 340 causes great regret - just here the change in the expression of the cell cycle regulators p16 and 21 is discussed, however, in the light of criticism of the data obtained on these proteins (see above comments to Fig. 4 and 5), there is no basis for the discussion itself until clarification results.
And it would be good to see the scheme of the described processes leading to opposite directions of compensatory responses in PE and IUGR against the background of centric insufficiency; for many readers, such a graphical generalization would be useful.

Author Response
Reviewer 2
The article of McCarthy and colleagues is devoted to the interesting problem of dissecting the mechanism of occurrence of various pathological manifestations of initial placental insufficiency in preeclampsia and/or IUGR. This problem is known and interesting because in both cases, oxidative stress in the placenta is involved in the development of the pathology, and the consequences are different - either just preeclampsia, or IUGR, or PE-associated IUGR. The issue of placental aging associated with the accumulation of oxidative damage has not yet been resolved, since data on aging at the morphological, molecular and functional levels are contradictory, despite the many studies in this area performed in the last decade. The authors set the task of showing whether cell aging, accompanied by shortening of telomeres, really determines the path of pathology development to varying degrees, or is it a matter of functional limitation through cell cycle inhibition?
Despite the fact that the authors obtained unequivocal data in favor of the second hypothesis (leave a small sample out of the brackets, the authors themselves understand that it would be nice to validate the study on an expanded sample), several lines of logic are mixed in the discussion of the results, which does not draw convincing conclusions.
For example, for what purpose the Аuthors in the chapter show the results that the absolute length of telomeres does not differ from the control anywhere (let's leave aside static tricks about values adjusted to age, etc. - this is unjustified for such small samples), and in the discussion Authors unexpectedly conclude –« Our results showed a significant decrease in senescence associated genes ATM, CCNB-1, PCNA and TBX2 in IUGR placentas, accompanied with shorter absolute placental telomere length and significant increase in maternal circulating Interferon-γ levels when compared to controls; whereas in pre-eclampsia, we showed a significant increase in placental expression of CHEK-1, CCNB-1, 264 PCNA, CDKN2A and PTEN, a significant reduction in p16 placental protein levels, extended absolute placental telomere length and a significant increase in maternal circulating IL-6 levels when compared to controls.
The authors have rephrased the discussion which now reads as follows:
“Our results showed a significant decrease in senescence associated genes ATM, CCNB-1, PCNA and TBX2 in IUGR placentas, and significant increase in maternal circulating Interferon-γ levels when compared to controls. Whereas, in pre-eclampsia, we showed a significant increase in placental expression of CHEK-1, CCNB-1, PCNA, CDKN2A and PTEN accompanied with a significant reduction in p16 placental protein levels, and a significant increase in maternal circulating IL-6 levels when com-pared to controls. There was no significant difference in in absolute placental telomere length between any study groups, when compared to controls.”
This is all the more incomprehensible, since the absence of a difference in telomere length against the background of functional shutdown of the cell cycle in IUGR is fully consistent with the Authors' concept of different paths for the development of a pathologyу from placental insufficiency to preeclampsia or IUGR.
What is the purpose of suddenly and without regard to the paradigm of the article mentioning the same content of mtDNA in the placenta in all groups and making a conclusion about the absence of mitochondrial dysfunction? The number of mtDNA copies is in no way related to the quality of mitochondria, and it is incorrect to speak of their dysfunction (rather than quantity) based on the definition of this parameter.
Taking into account other shortcomings listed below, the article should be finalized for subsequent publication, since the new data presented in it and the differences in the mechanism of pathology formation in PE and IUGR are of great importance for appropriate further research and possible clinical recommendations, and the contribution of the Authors is undoubted.
Questions and recommendations:
Abstract and introduction
- Although the Abstract lists the abbreviations of proteins, the content of which changes in the placenta and plasma during PE and IUGR in comparison with the control, nothing is deciphered in the Introduction (and even these proteins are not mentioned) - what are these proteins and why it is interesting to study these proteins, there is no logic, leading to the formulation of the problem.
The authors acknowledge the reviewer’s comment. The introduction has been updated to include the proteins. The manuscript now reads as following:
“Premature cellular senescence can be classified by activation of DNA damage response pathway and its’ associated signalling kinases which culminate in the stimulation of cell cycle inhibitors such as p16INK4a, p21/p53 and p27, reduction in cellular proliferation and loss of Lamin-B activity which is a marker of the altered enlarged and flattened morphology of senescent cells [9]. Moreover, the accumulation of mitochondrial DNA (mtDNA) as a result of ROS inducing damage and strand breaks, have been associated with premature onset of ageing in both in-vivo and in-vitro models, with higher susceptibility to mtDNA mutagenesis [10, 11]. Activation of a senescence associated secretory phenotype (SASP) incorporating a core set of cytokines such as IL-6, IL-8, IL-13, IGFBPs and MMPs result in accumulated senile cells creating a pro-inflammatory microenvironment that indicates establishment of senescence [12]. The phenotypes of premature cellular senescence are highly heterogeneous and often differ depending on the initial stressor [13]. Senescence specific gene expression has been studied extensively. Genes such as PTEN and CDKN2A regulate cellular senescence by acting through the p16/p19Arf pathway, whereas CHEK-1, PCNA and CCNB-1 are modulated by DNA damage response and halt in cell cycle progression [14-16]. Additionally, there is no single universal biomarker that specifically identifies senescence, hence a combination of multiple markers when present simultaneously, are used to confirm senescence [17].”
The selection of senescence associated genes were based on a preliminary experiment performed by the authors using the Qiagen RT2 Profiler PCR Array. The authors have mentioned this panel in the methods section.
- Again, why is mtDNA mentioned in terms of quantification and where is it supposed to see the significance of changes in the amount of mtDNA, especially as a result of damage (increase?) - in tissue, in plasma, in the intercellular space? "Moreover, the accumulation of mitochondrial DNA (mtDNA) as a result of ROS inducing damage and strand breaks..." - Probably, did the authors mean the accumulation of mtDNA MUTATIONS as a result of ROS? But further nowhere is it said about mutations and they were not defined. We also note that the vision of the Authors in the style of "...result from poor placental perfusion, resulting in ROS accumulation that subsequently..." is incorrect, since ROS cannot accumulate by themselves due to their high reactivity and short lifetime.
The authors have included mtDNA copy number quantification as a surrogate marker to assess the role of mtDNA‐mediated mitochondrial dysfunction in placental cellular senescence in adverse pregnancy outcomes such as PE & IUGR. The authors have justified the addition of mtDNA copy numbers in this publication, as a part of the discussion, as follows:
“Mitochondrial dysfunction has been associated with disease severity and complexity. Mitochondrial DNA (mtDNA) copy number is a measure of mtDNA levels per cell, is associated with mitochondrial enzyme activity and adenosine triphosphate (ATP) production. Mitochondria, as the source of ROS production, plays a pivotal role in pathophysiology of placental hypoperfusion disease as well as cellular dysfunction and ageing. Multiple publications have shown increased mtDNA in maternal circulation and cord blood, in pregnancies affected by PE, PE/IUGR and IUGR [50, 51]. Contrarily, the quality of mitochondrial quality and mtDNA copy number is significantly reduced in ageing cells [52-54]. Our results don’t provide evidence of increased mtDNA in PE, PE associated with IUGR and IUGR for mtDNA and based on this surrogate marker of mitochondrial dysfunction have proposed that mitochondrial dysfunction is not a significant contributor to the heterogeneity evident in our study.
- I would like to see a clearer logic here, for example (not necessarily exactly this way, I do not insist!) In the style - incomplete trophoblast invasion - permanent placental reperfusion - hypoxia / reoxygenation - oxidative stress - apoptosis and compensatory processes to restore placental function - such and such with PE, such and such with IUGR and the difference at the level of different regulation of tissue aging.
The authors would like to draw the reviewer’s attention to the introduction section detailing the placental senescence logic. The paragraph reads as follows:
“This stress-induced senescence is proposed to result from poor placental perfusion, resulting in increased ROS generation that subsequently engages biochemical pathways which disrupt cell cycle progression in addition to accelerating activation of proinflammatory cytokines and chemokines which form part of the Senescence-Associated Secretory Phenotype (SASP) [13,19].
In adverse pregnancy outcomes such as pre-eclampsia and IUGR, impaired placentation and the subsequent burden of placental oxidative stress may trigger a premature senescence phenotype leading to early parturition and pre-term delivery of the fetus [13]. Syncytiotrophoblasts show features characteristic of senescent cells including the biomarker Senescence Associated β-galactosidase (SA-β-gal), together with in-creased expression of the cyclin kinases inhibitors p16 and p21, and p53”
Results:
- Just mistake I expect phosphatase
137 line “…Phosphotase-Tensin homolog PTEN ….”
The authors have corrected the typographical error to “Phosphatase-Tensin homolog”.
- Fig 4 – it is not clear meaning of blots there – Is it typical blots? Should be written then in the caption to the figure.
Are these differences everywhere, almost an order of magnitude, between the samples of the control group? And why do the values for the control group vary so much between experiments - medians around 0.35 for a), less than 3 for b) and about 0.1 for c)? The nature of the distribution is also very different. Is there an explanation why tubulin-normalized values change so much for the same group (n=5) depending on what they are paired with? Generally unconvincing.
The authors would like to clarify that it is a typical SDS-page blot. The median value of protein expression for p16/B-tubulin ratio for each group is as follows:
Control |
PE |
0.35 |
0.15 |
Control |
PE/IUGR |
0.24 |
0.33 |
Control |
IUGR |
0.12 |
0.13 |
We have added the clarification to the figure caption, which now reads:
“Western Blot for placental p16 protein expression using standard SDS-Page method for (A) Pre-eclampsia only, (B) Pre-eclampsia associated with IUGR, and (C) IUGR. Protein expression of p16 was normalized to the expression of the housekeeping reference protein B-tubulin.”
The variation between the cases and controls can be explained by the fact that these are individual patient samples were age matched between cases and controls. The variations we see in the bands maybe a cause of individual protein variation between each study participant. Keeping in mind, these are human samples, these variations can be expected. Three separate placental control groups were used for the western blot experiment, no control was repeated in any group.
The authors have performed Wilcoxon matched-pair rank test and the decrease in p16 expression in PE was statistically significant for PE compared to controls (p=0.0313). There was no significant difference in p16 expression in PE associated with IUGR (p=0.0625) or IUGR (p=0.4) when compared to controls.
The authors have clarified this by adding the following in the methods section:
“PE, PE associated with IUGR and IUGR cases were age-matched with separate control groups”
- The values of the medians for p21/tubulin in Fig. 5 for control samples are even more depressing - 0.5; 0.3; 2.5 for a), b), c). How can it be in principle for samples of the same group, if we are discussing relative values (normalized)? - from blot to blot, p21 and tubulin stain better, sometimes worse, and this leads to a change in their ratio? what then does the definition and comparison with PE and IUGR, PE+IUGR matter? Maybe there is the result of an artifact? This should be analyzed while related samples and appropriate statistical method. Wilcoxon, for example, not MW!
The authors have reanalysed the p21 densitometry data for the IUGR study group and updated the results section
The authors found no significant difference in p21 expression in PE, PE associated with IUGR or IUGR only cases when compared to controls using the matched paired Wilcoxon Test. Similar to p16, samples were age matched between cases and controls. The variations we see in the bands maybe a cause of individual protein variation between each study participant. Keeping in mind, these are human samples, these variations can be expected. Three separate placental control groups were used for the western blot experiment, no control was repeated in any group. Moreover, the limited sample size maybe a contributing factor to the variation between the different phenotypes. The median value of protein expression for p21/B-tubulin ratio for each group is as follows
Control |
PE |
0.47 |
0.57 |
Control |
PE/IUGR |
0.34 |
0.32 |
Control |
IUGR |
0.201 |
0.26 |
The authors have also modified the methods section to update the statistical methodology used for WB experiments, as follows:
“Two tailed Wilcoxon matched-pairs signed rank test has been used for statistical analysis of Figure 4 and 5. P values <0.05 were considered as statistically significant.”
- Line 225 – “Senescence Associated Secretory Phenotypes (SASP) consists of a core panel of inflammatory cytokines and mediators including…” – where is this panel came from? Ref?
The authors have added references to identify the profile of the SASP panel in the manuscript. The references added are:
“24. D. Alimbetov et al., "Suppression of the senescence-associated secretory phenotype (SASP) in human fibroblasts using small molecule inhibitors of p38 MAP kinase and MK2," (in eng), Biogerontology, vol. 17, no. 2, pp. 305-15, Apr 2016, doi: 10.1007/s10522-015-9610-z.
- A. D. Hudgins, C. Tazearslan, A. Tare, Y. Zhu, D. Huffman, and Y. Suh, "Age- and Tissue-Specific Expression of Senescence Biomarkers in Mice," (in English), Frontiers in Genetics, Original Research vol. 9, 2018-February-23 2018, doi: 10.3389/fgene.2018.00059.
- J. P. Coppé, P. Y. Desprez, A. Krtolica, and J. Campisi, "The senescence-associated secretory phenotype: the dark side of tumor suppression," (in eng), Annu Rev Pathol, vol. 5, pp. 99-118, 2010, doi: 10.1146/annurev-pathol-121808-102144.”
A separate issue is the technique and quality of Western blots.
- First, it is not clear how two different signals from two proteins can be seen on the same blot (original images). Was the stripping technique used, was the overlay used in the program, why is this not described in the methods? And to my regret - Lamin B1 and GAPDH on the left and on the right are different membranes, not “same blot” - there the corners are cut off in different places and the markers dispersed in different ways on electrophoresis.
The stripping method was used on membranes to remove all antibody traces before probing with the housekeeper protein. The authors have added the stripping method to the main manuscript. The manuscript now reads:
“After the protein of interest was developed, membranes were stripped of residual primary and secondary antibodies using a Western Blot Stripping Buffer (Thermo Fischer, Ireland) for 30 minutes with gentle shaking and re-probed with a suitable housekeeper protein.”
The Lamin-B1 blots have been revised by the authors and reuploaded for review.
- Secondly, the methods do not specify which markers were used.
The authors have detailed the antibody markers used in western blots in the methods section.
“4.6. Protein Isolation and Western Blot
Placental tissue (50mg) was homogenized using TissueLyser II (Qiagen, Germany) and protein isolated using Tetraethylammonium bicarbonate buffer (Sigma-Aldrich, Ireland) supplemented with Phosphatase inhibitor (PhosSTOP™, Roche, Ireland) and Protease Inhibitor (Roche, Ireland). PE, PE associated with IUGR and IUGR cases were age-matched with controls. 35μg protein was separated by SDS-PAGE on a 10% TRIS polyacrylamide-glycine gel (Bio-Rad Laboratories, Ireland) and transferred to a methanol activated PVDF membrane (Amersham™ Hybond® Sigma-Aldrich, Ireland). Membranes were blocked in 5% BSA for 1 hour at room temperature and incubated overnight at 4oC with primary antibodies; anti-p21 (1:1000 Abcam; ab109520, UK), anti-Lamin B1 (1:1000 Cell Signaling; E6M5T, Ireland), anti-p16 (1:500Abcam; ab108349, UK), or anti-GAPDH (1:3000 Cell Signaling #2118, Brennan and company, Ireland) and anti- β-Tubulin (1:2000 Cell Signaling, #2146, Brenan and company, Ireland), secondary anti-rabbit IgG, HRP-linked antibody (1:2000 Cell Signaling Technology, #7074 Brenan and company, Ireland). GAPDH and β-Tubulin were selected as internal reference control proteins for western blots because of their stability and expression within placental cells [63-65]. After overnight incubation membranes were washed and incubated at room temperature for 1hr with anti-HRP secondary antibody, Membranes were washed in TBS-T and developed using Pierce™ ECL Western Blotting Substrate, Thermo Fischer Scientific, Biosciences, Ireland) and LICOR Odyssey image analyzer. Densitometry was performed using LICOR Image Studio Lite Version 5.2. After the protein of interest was developed, membranes were stripped of residual primary and secondary antibodies using a Western Blot Stripping Buffer (Thermo Fischer, Ireland) for 30 minutes with gentle shaking and re-probed with a suitable housekeeper protein.”
- Thirdly, in Supplementary Figure II, the band corresponding to this protein is quite clear, while in samples from the original images it is generally difficult to determine where the protein is localized, everything is so nonspecific (separation is generally unsuccessful).
The authors have uploaded the original blot images as part of this submission.
In general, after looking at the original images, it is not clear how such data on the content of these proteins were obtained, the blots are inconclusive.
Discussion:
Discussion of the results is a strong part of the article, although a description of the operation of individual proteins could be given in the introduction. Logical and persuasive.
- The only negative is long and poorly readable sentences (for example, from line 260 to line 267), but this is easy to fix.
The authors have revised the sentences from line 260-267. The following has been added to the manuscript:
“While physiological triggers of senescence in vivo are not fully understood, evidence of hypoperfusion coupled with high oxidative stress and placental insufficiency may orchestrate a multistep cellular senescence response. This study sought to assess the evidence of placental ageing resulting from cellular senescence in both Pre-eclampsia and IUGR by evaluating multiple biomarkers of senescence. Cellular senescence is defined as defined as a state of terminal proliferation, accompanied by characteristic metabolic and pro-inflammatory changes. Our results showed a significant decrease in senescence associated genes ATM, CCNB-1, PCNA and TBX2 in IUGR placentas, accompanied with shorter absolute placental telomere length and significant increase in maternal circulating Interferon-γ levels when compared to controls”
- The discussion from lines 320 to line 340 causes great regret - just here the change in the expression of the cell cycle regulators p16 and 21 is discussed, however, in the light of criticism of the data obtained on these proteins (see above comments to Fig. 4 and 5), there is no basis for the discussion itself until clarification results.
The authors have reanalysed and updated the result for p21 and p16 protein. Please see response to Reviewer 2’s comments summarized in points 2 & 3 in the results section of this letter.
- And it would be good to see the scheme of the described processes leading to opposite directions of compensatory responses in PE and IUGR against the background of centric insufficiency; for many readers, such a graphical generalization would be useful.
The authors have added the following graphical abstract for visualizing the differences between PE and IUGR phenotypes when compared to controls. The graphical abstract has been referenced in the main manuscript as well. Please find the image in the attached response letter.

Round 2
Reviewer 1 Report
all issues addressed
Reviewer 2 Report
The authors have done a great job and the result is deeply satisfying both in the semantic and qualitative aspects. It remains to wish a successful continuation of this interesting and important research.